# What matters most to older adults in treatment decision making: A discrete choice experiment

Vera C. Hanewinkel[1]*, Hanneke van der Wal-Huisman[2], Suzanne Festen[3], Richte CL Schuurmann[2], Goudje L. van Leeuwen[2], Maria-Annette Kooijman[2], Marijke J. Nogarede[2], Barbara L. van Leeuwen[2], Daan Brandenbarg[4]

1 Department of Policy, University of Groningen, University Medical Center Groningen, Groningen, The Netherlands, 2 Department of Surgery, University of Groningen, University Medical Center Groningen, Groningen, The Netherlands, 3 University Center for Geriatric Medicine, University of Groningen, University Medical Center Groningen, Groningen, The Netherlands, 4 Department of Primary and Long-term Care, University of Groningen, University Medical Center Groningen, Groningen, The Netherlands

* v.c.hanewinkel@umcg.nl

## Abstract

### Introduction

Medical decision making is often guided by disease-specific outcomes such as life extension or survival. Especially for older adults other outcomes like maintaining independence can be equally vital or more important. Enhanced insight into the priorities of community dwelling older adults can optimize treatment decision making and refine healthcare policy.

The aim of this study was is to identify which outcomes are prioritized by adults of 50 years and older when choosing between treatment options with various outcomes in a hypothetical case of a life-threatening disease.

### Methods

We conducted a Discrete Choice Experiment (DCE) with individuals aged ≥50, comparing six pairs of hypothetical treatment options with five attributes: life expectancy, independence, pain, memory complaints and societal costs. Attribute utility was analyzed using a conditional logit model, and latent class analyses were employed to explore preferences in groups. Data collection took place at a four-day national fair for people aged 50 or older in Utrecht, Netherlands.

### Results

In total 333 volunteers (233 female, mean 70 yr, SD 7.7) completed the DCE. Most participants prioritized maintaining independence, followed by life expectancy and the avoidance of severe physical or cognitive impairments (p < .05). Life expectancy only had a positive impact when it was extended by two years. Avoiding high societal costs also influenced preferences. Latent class analysis identified two

**Data availability statement:** All original raw data files (2x) and syntax (doFiles) (1x) are available in the Dutch national centre of expertise and repository for research data (DANS). Accession number (DOI): https://doi.org/10.34894/UIFAMX.

**Funding:** The author(s) received no specific funding for this work.

**Competing interests:** The authors have declared that no competing interests exist.

subgroups: one (approximately 25% of participants) prioritized life expectancy, willing to compromise on other factors, while the other emphasized independence and minimizing societal costs. Interaction tests between respondents' characteristics and preferences showed no significant differences.

## Conclusions

When confronted with a (possible) life threatening disease, most people aged ≥ 50 yr prioritize remaining independence and the absence of severe pain and severe memory complaints above life expectancy. A smaller subgroup prioritized life expectancy above other attributes. These attributes – including societal costs – should be taken in to account in individual treatment decision making, in clinical guidelines and national healthcare policy.

## Introduction

Technological, societal and medical innovations have extended the lifespan and wellbeing of individuals, contributing to an aging population. Life extension, however, does not always imply healthier individuals as older people often have more non-communicable diseases, which aggravates the burden on healthcare systems [1–3]. Western medicine, particularly in diseases with curative potential, relies on clinical trials based guidelines. These guidelines often focus on disease-centered outcomes, such as survival or life expectancy as primary outcome [4,5]. There is an increasing body of literature to support that for many older patients other outcomes are also important for instance functional outcomes like independence or balancing quality of life versus length of life. [6–11]. What matters most to them is often driven by their values in life. A thorough understanding of a patient's situation and preferred outcomes helps to align treatment choices more effectively to patients' values with equal or better outcomes [6,9,12]. Interestingly, studies have shownthat healthcare professionals are often unaware of their patients' treatment goals [13–15]. Consequently, treatment decision making often relies on treatment options according to guidelines, but these may not always achieve desired outcomes for older patients undergoing these treatments.

Although studies have been performed on treatment preferences among patients that need to make an actual treatment decision or patients with chronic diseases, there is limited data on how a general population of older adults weigh these outcomes in treatment decisions and what matters most to them when confronted with incompatible outcomes [13–15] This lack of knowledge contributes to an ongoing focus on life extension, which may result in care that is not well-suited to the individual patient (overtreatment)thereby putting additional pressure on already limited healthcare resources [16].

While treatment goals and preferred outcomes are known to be important factors in decision making for patients government bodies also consider the societal costs in their evaluations of health care programs [17]. Over the past decades the

increase in demand for healthcare and limited resources in terms of labor and finance has led to scarcity and pressure on the current healthcare system [18]. This scarcity has increased interest in healthcare choices and values of individuals, which is a crucial aspect of designing policies that accurately reflect society's desires [19]. So far, it is unknown if individuals in the Dutch healthcare system, in which healthcare expenses are funded through regulated and mandatory health insurances with limited out of pocket payments from patients, consider societal costs an outcome in treatment decision-making.

With this research we aim to gain insight into the preferences and prioritization regarding treatment outcomes of older adults in a general population. Increased knowledge about preferences of older adults with regards to treatment outcomes can contribute to personalized treatment decision making for individuals, but it also urges the need for a public debate on an exhausting healthcare system that is currently mainly focused on survival [3,20]. It is particularly relevant to gain insights from this group of 'soon-to-be' healthcare consumers, as they are likely to contribute significantly to the healthcare burden in the coming decade. Gaining insight into these preferences can support evidence-based policymaking and guide the prioritization of meaningful outcomes in both clinical guideline development and future clinical trials.

Therefore, the aim of this study is to explore which treatment outcomes are prioritized by people over 50 years of age when they get to choose between two treatment options with various outcomes, in a hypothetical scenario of life-threatening disease.

## Methods

### Discrete choice experiment

We performed a cross-sectional survey using a Discrete Choice Experiment (DCE). A DCE is a survey-based statistical technique used to understand preferences by presenting individuals with a set of hypothetical choices among two scenarios. Each scenario is characterized by several attributes, and each attribute has different levels. Levels refer to the different variations that each attribute can take. Participants were presented with a series of hypothetical choice tasks consisting of two or more scenarios and were asked to make a choice for the scenario they preferred. In this case, the scenarios consisted of hypothetical treatment options for a potentially life-threatening disease for which a treatment decision could be made, with various kinds of treatment outcomes as attributes. By analyzing the choices, the relative importance of each attribute could be identified, together with the preferred levels within those attributes.

The method is based on Lancaster's theory of value and McFadden's Random Utility Theory [21,22], and assumes that all attributes have a certain utility. Utility is a subjective measure to indicate how much a person values a specific attribute. People are assumed to choose the scenario with the highest utility for them, so that gives them the most value. [21–24]

### Attributes and levels

The scenario was presented as a hypothetical life-threatening disease for which a treatment decision could be made, without further specification, allowing respondents to interpret it within the context of their own situation. At the end of the survey, respondents were asked whether they had a specific disease in mind while answering the questions and, if so, to indicate which one. The exact wording of the scenario is presented in Box 1. The attributes and levels for the current DCE were derived from literature and compiled in multiple meetings with the research group consisting of healthcare professionals with a track record on personalized care, shared decision making, geriatrics, surgery and research methodology.

Based on literature and experience in determining preferences and goals, the Outcome Prioritization Tool (OPT) was chosen as initial framework for the compilation of the attributes [8,14,25]. The OPT consists of four universal health outcomes: life extension, maintaining independence, reducing or eliminating pain and reducing or eliminating other

complaints [26]. Starting from this framework, alterations were made. Life extension was kept the same, as was maintaining independence. We adjusted 'the reduction of pain' into 'having pain or physical complaints' to broaden the subject. The reduction of 'other complaints' was adjusted in to 'having memory complaints' as this is a relatively common problem among older adults which can substantially affect daily function [27]. An attribute on costs of treatment for society was added to the experiment. This is a relevant attribute in the public debate as health expenditures are rising in the Netherlands and both financial and workforce scarcity are an increasing problem [17–19,28]. In the Netherlands every citizen is obliged to have healthcare insurance with insurance premium. This attribute is defined as the costs of the treatment, paid by the insurance or the government. An overview of the attributes and levels is provided in Table 1.

---

### Box 1. Scenario presented to participants

*Imagine the following: you are seriously ill. The specific illness is not relevant here. The illness is severe enough that it could lead to death.*
*A treatment is available for this illness.*
*In this study, you will be presented with two treatment options each time: Option A and Option B.*
*Each option is described by several characteristics. These indicate the expected outcomes of the treatments.*
*Please choose the option you prefer in each scenario.*
*If you feel that neither option is suitable, we still ask you to select the one you consider most appropriate.*

---

All attributes consisted of three levels. The division of levels was based on literature and expert opinion of the team. In the draft version *Life extension* was first categorized into one year, five years and normal life expectancy. *Remaining independence* was divided into doing everything yourself, being dependent on others (family or home care) and living in a nursing home. Pain or physical complaints and memory complaints were divided into no complaints, mild complaints or severe complaints. Societal costs were divided according to the Dutch National Care authority. These cut-off scores are based on care burden in relation to the monetary value of a quality-adjusted life year (QALY). In this study, we translated this into treatment costs, categorized as low (€20,000), medium (€50,000), or high (€80,000) per treatment [17].

**Table 1. Attributes and levels of the experiment.**

| Attributes | Levels | Explanation for participants |
|---|---|---|
| Life extension | • 6 months<br>• 2 years<br>• 5 years | 'how long you will live with the treatment'. |
| Independence | • Being able to continue to do as much as possible yourself<br>• Being dependent on others (for example home care or informal care)<br>• Living in a nursing home | 'how much can you still do independently after the treatment'. |
| Pain or physical complaints | • no pain or physical complaints<br>• mild pain or physical complaints<br>• severe pain or physical complaints | 'amount pain or other complaints you will have after the treatment' |
| Memory complaints | • No memory problems<br>• mild memory problems<br>• severe memory problems | 'how much memory problems you will have after treatment' |
| Costs to society | • Low: 20,000. Per treatment<br>• Medium: 50,000 per treatment<br>• High: 80,000 per treatment | 'how much does the treatment cost to society' |

To enhance understanding of final attributes and levels, we obtained feedback from an independent group of five older adults who were not involved in the study. Members of the research team invited these adults to test the model and discussed their findings in face-to-face meetings. Based on these sessions we adjusted the levels of some of the attributes: Life expectancy was divided into six months, two years and five years, which are common cut-off points in literature and clinical practice [29]. Also, respondents indicated that the levels should be close enough to make it a hard decision but with a relevant range and this indicates that the illness is life threatening. The attribute of independence was adjusted on the first level to 'being able to continue to do as much as possible yourself'. The independent group indicated that complete independence was deemed unlikely under these circumstances. All choice tasks were composed in a way that the options are clinically possible, and unlikely combinations were excluded [30].

Treatment options were not further defined; participants were invited to provide their own interpretation. At the end of the questionnaire respondents were asked whether they had a specific illness in mind and if so to report the illness in the questionnaire.

## Design

A full factorial design with all 243 scenarios (3^5) was not feasible. Instead, a fractional-factorial design was used, selecting a subset of combinations to maximize information on main effects and key interactions (D-efficiency) for a conditional logit model [27,28] in Ngene 1.3 (ChoiceMetrix). Bayesian priors were estimated in the experimental design stage based on prior clinical data, to construct a Bayesian D-efficient design [31]. The priors reflected expected directions of effects based on existing clinical evidence (e.g., negative for losing independence, positive for life expectancy). A pilot study (30 respondents) assessed understanding, burden, and willingness to participate. The pilot data was included in the final dataset.

To estimate the required sample size, we applied the rule of thumb proposed by Johnson and Orme [32,33] for conjoint/discrete choice experiments: $N \geq \frac{500 \times c}{t \times a}$.

In this equation c is the largest number of levels for any attribute, t is the number of choice tasks per respondent, and a is the number of alternatives per task. In our design, the maximum number of attribute levels was three, each participant completed six choice tasks, and each task contained two alternatives. Thus, according to this rule of thumb, a minimum of 125 respondents would be required for reliable estimation of main effects. The experiment included 18 choice tasks, split into three blocks to reduce participant burden. The blocked design ensured that each scenario was observed at least 15 times across respondents, providing sufficient variation for reliable estimation. The questionnaire was created in Qualtrics (Qualtrics, Provo, Utah), with an example shown in Fig 1.

## Recruitment of participants and data collection

Participants were volunteers aged 50 + recruited at a four-day national fair (12th – 16th of September 2023) in Utrecht, The Netherlands. They had to read and understand Dutch and provide informed consent based on verbal and written information on an electronic device (tablet). No further exclusion criteria applied.

| Attribute | Treatment A | Treatment B |
|---|---|---|
| Life expectancy | 2 Years | 6 months |
| Independence | Living in a nursing home | Being able to continue to do as much as possible yourself |
| Pain or physical complaints | Mild pain or physical complaints | Mild pain or physical complaints |
| Memory complaints | No memory problems | Mild memory problems |
| Costs to society | High: 80,000 euros per treatment | Low: 20,000 euros per treatment. |

**Fig 1. example of choice task.**

After a brief introduction, participants received a tablet with the choice experiment. They provided informed consent and personal details, including gender, age, four-digit postal code, and education level. They also indicated their living situation (independent, with informal/home care, or in a nursing home) and rated their health status on a 0–100 visual analogue scale [34].

To facilitate participation, chairs, reading glasses, and an instruction sheet were available. Researchers or a relative assisted if needed. The questionnaire, with mandatory fields and no opt-out, was created using Qualtrics (Qualtrics, Provo, Utah).

### Statistical analysis

Descriptive statistics were performed to describe background characteristics of the included sample. Mean age and median health status were calculated. Educational level was assessed and divided into three categories, primary education, lower secondary education, and higher secondary education/university level [35].

A conditional logit model for the main effect of the attributes was derived. All attributed were treated as categorical variables with dummy-coding. The utility of all attributes was individually tested using a Wald chi square test.

In addition to the main effects, a set of interaction terms between selected respondent characteristics and choice attributes was specified. These interactions were determined *a priori*, informed by theoretical considerations, previous literature on heterogeneity in preferences, and consensus within the study team [36–38]

Interaction testing was not included in the main model, but – based on literature and consensus of the study team – specific pre-specified interactions between attributes and personal characteristics (gender and physical pain, age and maintaining independence, educational level and societal costs, gender and independence) were tested using Wald tests.

Further, data was analyzed using latent class analysis. Latent class analysis is a model-based clustering approach that identifies unobserved (latent) subgroups within the sample, based on similarities in choice patterns. This method allows for the estimation of distinct classes of respondents with relatively homogeneous preferences, thereby providing insights into systematic differences in decision-making across subpopulations., The number of classes was determined by the lowest consistent Akaike information criterion (CAIC). Class memberships were analyzed to define characteristics of the class members. All data was analyzed using STATA 17 (StataCorp, Release 17. Texas).

The medical Ethics Review Board of the University Medical Center Groningen reviewed the research protocol and concluded that the protocol is not a clinical research with human subjects as meant in the Medical Research Involving Human Subjects Act (WMO) (Ref nr M23.320013).

## Results

### Study population

A total of 337 participants were recruited. Four participants did not complete the experiment and were therefore excluded, leaving 333 respondents in the analysis (Table 2).

Participants had a mean age of 70 (±7.7), most participants were female (70.0%), and lived independently (96.1%). Participants reported a median health status of 80 (17.5). Most participants completed lower secondary education (60.7%).

### Patients' preferences

The conditional logit model showed significant utility on all attributes (p < .05), but not on all levels (Table 3). Positive utility was found for life expectancy, but only for a two-year increase (0.46, 95% CI 0.29 to 0.62, p < 0.05), a five-year increase showed no significant utility (.17, 95%CI −.02 to.36, p = .09). Maintaining independence showed significant negative utility on both living in a nursing home and being dependent on others (−.76, 95% CI −.92 to −.61, p < 0.05) (-.1.19, 95% CI −1.36 to −1.02, p < 0.05) where physical pain and memory complaints only showed significant negative utility for severe complaints (−.82, 95% CI −.99 to −.64, p < 0.05) (−1.16, 95% CI −1.36 to −.96, p < 0.05).

**Table 2. Participant characteristics.**

| Patient characteristics | Total N 333 (%) |
|---|---|
| **Age (years) (mean, SD)** | 70 (7.7) |
| **Sex** | |
| *Female* | 233 (70.0) |
| *Missing* | 1 (0.3) |
| **Living situation** | |
| *Independent* | 320 (96.1) |
| *Independent with informal care or home care* | 10 (3.0) |
| *Nursing home* | 2 (0.7) |
| *Missing* | 1 (0.3) |
| **Health status (median, IQR)** | 80 (17.5) |
| **Education** | |
| *Primary education* | 15 (4.5) |
| *Lower secondary education* | 202 (60.7) |
| *Higher secondary education/ university level* | 114 (34.2) |
| *Missing* | 2 (0.60) |

**Table 3. Conditional logit model DCE.**

| Variables | | COEFFICIENT | P | 95% CONF INTERVAL |
|---|---|---|---|---|
| **Life expectancy** | **Reference: 6 months** | | | |
| | 2 years | .456 | .000* | .287 –.624 |
| | 5 years | .166 | .087 | −.024 –.356 |
| **Independence** | **Reference: as independent as possible** | | | |
| | Dependent on others (home care/ informal care) | −.764 | .000* | −.924 – −.605 |
| | Living in a nursing home | -.1.189 | .000* | −1.363 – −1.017 |
| **Pain or physical complaints** | **Reference: no pain or physical complaints** | | | |
| | Mild pain or physical complaints | .144 | .166 | −.060–348 |
| | Severe pain or physical complaints | −.816 | .000* | −.991 – −.641 |
| **Memory problems** | **Reference: no memory problems** | | | |
| | Mild memory problems | −.031 | .753 | −.227 –.164 |
| | Severe memory problems | −1.160 | .000* | −1.357 – −.963 |
| **Societal costs** | **Reference: Low: 20,000 euros** | | | |
| | Average: 50,000 euros | −.172 | .011* | −.304 – −.039 |
| | High: 80,000 euros | −.174 | .038* | −.339 – −.010 |

* p < .05

All attributes were found to be relevant for decision making (p < .05) with independence being most impactful (1.19). An overview of the results is presented in Fig 2.

## Preferences and personal characteristics

Testing between respondents' characteristics and specific attributes showed no significant differences. Severe physical pain and living in a nursing home showed no significant difference for men compared to woman (p = .420), (p = .426). Living in a nursing home showed no significant difference for older (>70yr) people compared to younger people (<70yr)

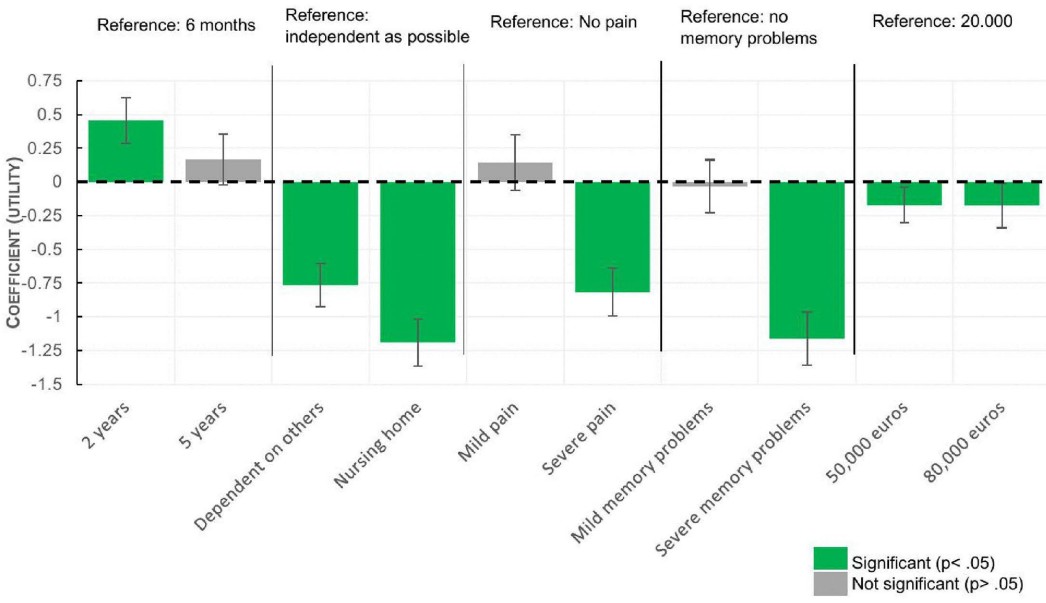

**Fig 2. attribute utility with confidence intervals.**

(p.681) and there was no significant difference found for high societal costs showed for people with a longer education compared to a short education (p = .181).

## Latent class analysis

Latent class analyses identified two groups, representing 22.4% and 77.6% of the respondents, respectively (Table 4). The percentages represent the average probability of respondents belonging to a class.

Class 1 represented a higher percentage men compared to class 2 (36.1% vs 28.0%), was younger of age (68.9 vs 70.45) and had a better health status (80.1 vs 76.9). People with a long education were relatively more represented in class one, compared to class two (42.3% vs 32.3%).

These classes differed in choice preferences on three levels: five years life expectancy, living in a nursing home and high societal costs. These are highlighted in Table 5.

Class one valued 5 years life expectancy more (p = .019 vs.940), whereas class two negatively valued living in a nursing home (p = .058 vs. < .001) and high societal costs (p = .053 vs. < .001) more than class one. The differences between the classes are highlighted in Fig 3.

**Table 4. *Characteristics of class members.***

| Sex | Class 1 | Class 2 |
| --- | --- | --- |
| Male | 36.1% | 28.0% |
| Female | 63.9% | 72.0% |
| Age | 68.9 | 70.45 |
| Health Status | 80.1 | 76.9 |
| Short education | 4.2% | 4.6% |
| Middle long education | 53.5% | 63.1% |
| Long education | 42.3% | 32.3% |

Table 5. Priorities of classes.

| Variable | Class1 Coefficient | p value | Class2 Coefficient | p value |
|---|---|---|---|---|
| Life Expectancy: 2 years | 1.621 | .005 | 0.324 | .007 |
| Life Expectancy: 5 years | 1.080 | .019 | 0.010 | .940 |
| Independence: care from others | −1.627 | .036 | −0.826 | .000 |
| Independence: Nursing home | −0.961 | .058 | −1.498 | .000 |
| Physical pain or discomfort: Mild | 0.666 | .253 | 0.103 | .546 |
| Physical pain or discomfort: Severe | −3.029 | .000 | −0.623 | .000 |
| Memory problems: Mild | −0.992 | .166 | −0.160 | .287 |
| Memory problems: Severe | −2.148 | .008 | −1.416 | .000 |
| Societal costs: Medium | −0.678 | .053 | −0.190 | .078 |
| Societal costs: High | 1.216 | .053 | −0.400 | .000 |
| **Class Share** | **0.224** | | **0.776** | |

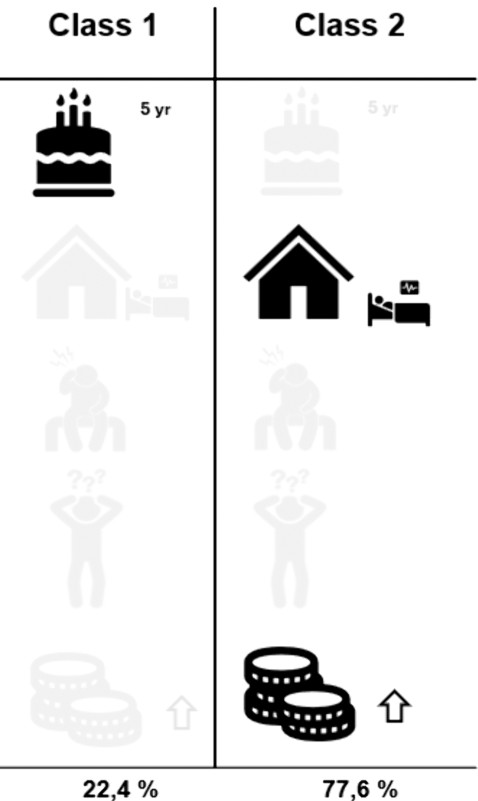

**Fig 3. Priorities of classes.**

## Illness in mind

After completion of the questionnaire, respondents were asked if they had an illness in mind when answering the questions. 78 respondents (23.4%) reported they had one or more illnesses in mind. Most mentioned by respondents are neurological disorders (33 times) followed by oncological disorders (32 times). 255 (76.6%) respondents reported they did not have an illness in mind. An overview of the illnesses mentioned are reported in Table 6.

**Table 6. Illness in mind.**

| Type of illness | Nr of times mentioned |
|---|---|
| Neurological Disorders[a] | 42 |
| Oncological Disorders | 32 |
| Musculoskeletal Disorders | 6 |
| Cardiovascular Disorders | 2 |
| Respiratory Disorders | 1 |
| Endocrine & Metabolic Disorders | 1 |
| Renal Disorders | 1 |
| Unspecified | 1 |

[a]: including Alzheimer's disease, Parkinson's disease, Ischemic stroke and brain injury.

## Discussion

This study is the first to explore prioritization of treatment outcomes, including societal costs, in a general population of people aged 50 years and older. The study reveals that maintaining independence was most important, but other factors such as life expectancy, the avoidance of severe physical and cognitive impairments are relevant as well. Another relevant finding was that for this study population, avoiding medium or high societal costs were also found to be a relevant factor that older people consider in decision making.

These preferences were consistent across most participants, although latent class analysis revealed two distinct sub-groups with notable differences in attribute prioritization. One class, comprised of roughly a quarter of the participants, prioritized life expectancy and appeared to be willing to compromise on other factors to gain additional years of life, whereas the other group values independence and avoiding high societal costs.

### Comparison with other studies

Our findings align with previous research with patients in a hospital setting, showing that, in general, older adults prioritize independence and quality of live over survival as preferred outcome [10,11,13,39]. Our research shows that minimizing the risk of living in a nursing home and avoiding severe physical or memory problems are key considerations in treatment decision making for older adults. In contrast, mild memory issues or mild physical complaints appeared to be more acceptable to older adults, if they do not come at excessive costs—whether in terms of health outcomes or societal expenses.

Our finding that 77% of participants placed less emphasis on life expectancy compared to maintaining independence is particularly noteworthy. This challenges the traditional focus on health outcomes, such as survival, which often shapes the development of new treatments and clinical guidelines. In an era marked by fast improvements in medical technology and rising healthcare expenses, this mismatch may not only contribute to rising costs due to overtreatment but also to undesirable outcomes for a growing number of older adults [40,41]. As we showed in our study, current guidelines may only suit a quarter of the population of older adults, specifically those that value survival over everything else [4,5]. For the majority, however, a more balanced approach is essential [42].

A noteworthy finding of this study is the role of life expectancy in choice patterns of respondents. While a two-year increase in life expectancy appeared to be relevant, an additional five years did not demonstrate the same influence. A possible explanation, supported by participants' comments, is that the perceived quality of these additional years plays a critical role. Participants may prefer a shorter life expectancy if the accompanying health outcomes or other treatment attributes are considered less desirable.What also stands out in this research is our finding that avoiding extensive societal costs were also consideredin decision making. Where prior research showed that individuals weigh 'out of pocket' costs

in their decision making, this research is the first to show that people care about the societal expenses of a treatment [43]. This finding creates an opportunity to open the conversation about healthcare costs and address them openly.

## Implications for practice and policy

These results give rise to recommendations for healthcare practice and policy. First, in healthcare, patient consultations should incorporate a more individualized and holistic approach, aligning with the principles of shared decision-making, which is the core value of patient-centered care [44]. While this study provides insights into the general priorities of older adults, it also highlights significant differences between groups. Since it is difficult to predict to which group an individual would belong –prioritizing independence or prioritizing life extension -, explicitly discussing treatment preferences remains essential, as healthcare professionals cannot otherwise accurately determine what an individual values most in their care [13]. Structurally incorporating these conversations into the decision-making processes is crucial. Second, policymakers should focus on measures to evaluate hospitals and other care facilities in a way that corresponds to the values of society and ideally find a way to finance a patient centered healthcare system accordingly. Implementing these recommendations could shift the healthcare system towards more patient-centered care and facilitate care that brings value to individuals and society.

As a final point, societal costs of medical treatment should be more often discussed in a public debate. They should not necessarily be discussed between doctor and patient, but both medical professionals and individuals should have increased knowledge and awareness of healthcare expenses and potential limitations of a healthcare system.

## Strengths and limitations

A key strength of this study is that we conducted this study in a general population of middle-aged and older adults in the Netherlands. In the Netherlands over 30% of inhabitants is 55 years or older and the population keeps on ageing [45]. The study population reflects the demographics of the Netherlands in terms of educational level and living situation, with 92% of people over 75 years old living independently at home [45,46]. The design of the study is a particularly adequate way to explicate the process of decision making, something that often remains implicit. To our knowledge this study is the first to take societal costs of treatment into account, which gives a fair representation of the public opinion on this topic in the Netherlands.

A limitation of this study is that participants were recruited based on their attendance at the fair and their interest in healthcare, potentially causing selection bias and limiting the generalizability of the findings. However, the specific fair is a national fair with people attending from all over the country. This makes it relatively generalizable for the population and probably one of the most diverse general populations specifically for older adults in this country. Where these results are now limited to the Dutch population, it would be highly interesting to repeat this study in other countries with similar care systems, as cultural values may cause a difference in priorities in other countries. This could bring valuable insights into priorities in a larger and more diverse population.

The choices in this study were based on hypothetical scenarios and people their response to it at that specific moment. Treatment preferences can shift over time [47], and it is possible that participants might prioritize different factors when they actually face a treatment decision. When asked whether respondents had an illness in mind, 23% stated they had something specific on their mind, but the reported illness were divers. The other 77% of respondents reported not having an illness in mind, but were able to answer the questionnaire without any problem. This suggests that respondents are capable handling the hypothetical scenario. The results from these hypothetical scenarios highlight the importance of discussing this topic within society at large and addressing key societal questions related to healthcare policy. Additionally, similar results have been found in studies involving real patients, suggesting that the preferences expressed in hypothetical situations may still be reflective of real-world decision-making [10,11,15]. Although we did not specifically ask participants about their current health status or ongoing treatments, it is important to note that with roughly 50% of all cancer

diagnoses appearing in individuals over the age of 50, the scenarios presented may not have been entirely hypothetical for all participants [48]

## Conclusion

When confronted with a scenario of possible life-threatening disease, for most people aged 50 years or older maintaining independence was the most important, but avoiding severe pain and memory complaints, increasing life expectancy and even avoiding high societal costs mattered in treatment decision making. Only a minority of people prioritized life extension over almost everything else. Both in practice and in policy all these attributes should be considered in individual treatment decision making, in new clinical guidelines and national healthcare policy.

### Key points

- When prioritizing treatment outcome, the majority of community-dwelling adults over 50 years of age prioritized maintaining independence but other attributes, including avoiding high societal costs, are relevant for decision making as well.

- A subgroup, comprised of roughly a quarter of the participants, prioritized life expectancy and appeared to be willing to compromise on other factors to gain additional years of life.

- Healthcare and policy should explicitly discuss treatment preferences to ensure valuable care for individuals and society.

## Author contributions

**Conceptualization:** Vera Hanewinkel, Hanneke van der Wal-Huisman, Suzanne Festen, Richte C L Schuurmann, Goudje L van Leeuwen, Maria-Annette Kooijman, Marijke J Nogarede, Barbara L van Leeuwen, Daan Brandenbarg.

**Data curation:** Vera Hanewinkel, Hanneke van der Wal-Huisman, Suzanne Festen, Richte C L Schuurmann, Goudje L van Leeuwen, Maria-Annette Kooijman, Marijke J Nogarede, Barbara L van Leeuwen, Daan Brandenbarg.

**Formal analysis:** Vera Hanewinkel, Richte C L Schuurmann, Barbara L van Leeuwen, Daan Brandenbarg.

**Investigation:** Vera Hanewinkel.

**Methodology:** Vera Hanewinkel, Richte C L Schuurmann, Daan Brandenbarg.

**Project administration:** Vera Hanewinkel.

**Software:** Daan Brandenbarg.

**Supervision:** Barbara L van Leeuwen, Daan Brandenbarg.

**Validation:** Vera Hanewinkel, Richte C L Schuurmann.

**Visualization:** Vera Hanewinkel.

**Writing – original draft:** Vera Hanewinkel, Hanneke van der Wal-Huisman, Barbara L van Leeuwen, Daan Brandenbarg.

**Writing – review & editing:** Hanneke van der Wal-Huisman, Suzanne Festen, Richte C L Schuurmann, Goudje L van Leeuwen, Maria-Annette Kooijman, Marijke J Nogarede, Barbara L van Leeuwen, Daan Brandenbarg.

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
