## [Decision Letter · Decision Letter 0]

2 Sep 2025

Dear Dr. Hanewinkel,

Thank you for submitting your manuscript to PLOS ONE. After careful consideration, we feel that it has merit but does not fully meet PLOS ONE’s publication criteria as it currently stands. Therefore, we invite you to submit a revised version of the manuscript that addresses the points raised during the review process.

We look forward to receiving your revised manuscript.

Kind regards,

José Alberto Molina

Academic Editor

PLOS ONE

Journal Requirements:

Reviewers' comments:

Reviewer's Responses to Questions

**Comments to the Author**

1. Is the manuscript technically sound, and do the data support the conclusions?

Reviewer #1: Partly

Reviewer #2: Yes

2. Has the statistical analysis been performed appropriately and rigorously?

Reviewer #1: Yes

Reviewer #2: Yes

3. Have the authors made all data underlying the findings in their manuscript fully available?

Reviewer #1: No

Reviewer #2: No

4. Is the manuscript presented in an intelligible fashion and written in standard English?

Reviewer #1: Yes

Reviewer #2: No

Reviewer #1: This paper employs a discrete choice setup to study older adults’ preferences when making treatment decisions. The manuscript addresses a relevant research question, and the overall study design is adequate to investigate it. That being said, there are some important limitations that should be addressed before the manuscript can be considered for publication. I recommend to offer the authors a revise and resubmit, and encourage the authors to carefully address the concerns raised in this review.

1. Data availability: I am somewhat confused with whether the data supporting these findings will be made public upon publication. In response to the question, “Do the authors confirm that all data underlying the findings described in their manuscript are fully available without restriction?” the authors wrote: “No – some restrictions will apply.” However, they also state that “data will be held in a public repository, and will be made available after acceptance”. Why not deposit the data in a public repository now and include the link in the submitted paper? Doing so (along with providing the code used in the analyses) would have allowed us to verify the specific analytical setup and make a more comprehensive evaluation of the work.

2. Statistical power: The authors state that “Each respondent completed six tasks with two scenarios, ensuring each scenario was tested at least 15 times”. What is the justification for this minimum of 15 observations per scenario? What a priori power analysis was conducted to determine the required sample size (e.g., assumed effect sizes, desired power, alpha level, and model structure)? Please report the power analysis and indicate whether the achieved sample meets those requirements.

3. Pilot: The pilot data was included in the final dataset. This is not a standard practice. Were there any differences between the design/respondents of the pilot and the main study?

4. In the study, each participant made six choices. Choices made by the same participant are not independent observations. To draw valid inferences, the authors need to account for this lack of independence in their logistic analyses (for example, by clustering standard errors at the participant level or using an appropriate multilevel model).

5. On page 7, the authors state that “Bayesian priors were estimated based on prior clinical data.” Could the authors clarify what these Bayesian priors were used for? Please, also specify the actual priors employed and run a sensitivity check.

6. The authors include certain interaction terms between personal characteristics and choice attributes in their models. In page 8 they argue that the specific interactions to be tested were chosen based on “literature and consensus of the study team”, without providing further explanation on these statistical design choices. The authors should provide a more detailed justification of these analytical choices, referencing relevant literature.

7. Not every reader will have a solid grasp on how latent class analysis works. The paper would benefit from providing a more extensive introduction to it.

8. When running the latent class analysis, was the unit of analysis the individual or the choice? The text seems to suggest that individuals were classified into two groups, but I am not convinced. If the unit of analysis is indeed the individual, and N=333 (as reported in Table 2), then there cannot be a group with exactly 22.4% of individuals, since this would correspond to 74.5 individuals. The group should consist of either 74 individuals (22.2% of the sample) or 75 individuals (22.5%). Could the authors clarify how the multiple observations per individual were handled in this analysis and how group percentages were derived?

9. What do the authors make of the non-monotonic life expectancy preferences? (the fact that people assign a higher utility to 2 additional life expectancy years than to 5 years). What could explain these findings? I think the authors should elaborate on this further in the discussion.

Reviewer #2: I find the manuscript to be timely and relevant. However, several issues need to be addressed before the paper can be considered for publication:

Language and clarity: The manuscript would benefit from careful proofreading and language editing to improve readability and ensure clarity of expression.

Presentation of Logit estimates: The current tables report coefficients without clearly indicating their interpretation. I recommend presenting average marginal effects, which would make the results more interpretable and policy-relevant.

Sample representativeness: While the Netherlands is indeed experiencing an aging process, the sample appears to cover a specific geographic area and may not be representative of the broader population. The authors should discuss potential sample selection issues more transparently and clarify the limitations of generalizing the results.

Addressing these points would strengthen the manuscript and improve its contribution to the literature.

**Do you want your identity to be public for this peer review?** For information about this choice, including consent withdrawal, please see our Privacy Policy

Reviewer #1: No

Reviewer #2: No

---

## [Author Response · Author response to Decision Letter 1]

1 Oct 2025

Reviewer 1

1. Comment 1; Data availability: Why not deposit the data in a public repository now and include the link in the submitted paper?

Response: We apologize for the lack of clarity in our initial statement. To address this concern, we will deposit the full dataset and the analysis code in a public repository prior to publication. A link to the public repository (Dutch national centre of expertise and repository for research data (DANS)) is included below and in the data availability statement to ensure full transparency and reproducibility.

https://doi.org/10.34894/UIFAMX.

2. Comment 2; Statistical power: What is the justification for this minimum of 15 observations per scenario? What a priori power analysis was conducted to determine the required sample size (e.g., assumed effect sizes, desired power, alpha level, and model structure)? Please report the power analysis and indicate whether the achieved sample meets those requirements.

Response: We acknowledge the reviewer’s request for a traditional a priori power calculation. However, such calculations are not standard practice in DCE methodology, as statistical power depends not only on sample size but also on design efficiency, attribute level balance, and the true (a prior unknown) parameter values. Instead, the literature recommends rules of thumb, most widely the Johnson & Orme formula [1, 2], which indicates the minimum sample size needed to reliably estimate main effects. Based on this approach, our sample size exceeded the recommended threshold. In addition, by ensuring that each scenario was observed at least 15 times, we safeguarded sufficient variation across attributes for model estimation. This approach is consistent with ISPOR task force guidelines[3]. We have clarified this in the Methods section (line 154-162).

3. Comment 3; Pilot: The pilot data was included in the final dataset. This is not a standard practice. Were there any differences between the design/respondents of the pilot and the main study?

Response: We agree that in some study designs pilot data are typically excluded from the main analysis. In DCE research, however, pilot studies are commonly used both to test comprehensibility and to generate parameter priors for an efficient design [1-4]. Because our pilot respondents were recruited using the same procedures and eligibility criteria as the main sample, and because the pilot design did not differ from the final experimental design, it is considered appropriate and standard practice to include these data in the final analysis.

4. Comment 4; Independent observations: In the study, each participant made six choices. Choices made by the same participant are not independent observations. To draw valid inferences, the authors need to account for this lack of independence in their logistic analyses (for example, by clustering standard errors at the participant level or using an appropriate multilevel model).

Response: In discrete choice experiments, each participant indeed makes multiple choices. The conditional logit model explicitly accounts for this by modelling the probability of each choice conditional on the available choice set, with the likelihood function defined over all choices of an individual [5]. Thus, the repeated-choice structure is already incorporated in the estimation procedure, and clustering of standard errors at the participant level is not required. To additionally explore preference heterogeneity across respondents, we performed a latent class analysis after the conditional logit estimation. This approach allows for the identification of subgroups with distinct preference patterns, complementing the main effects estimated by the conditional logit model. Together, these analyses are in line with best practices in DCE methodology [3].

5. Comment 5; Bayesian priors: On page 7, the authors state that “Bayesian priors were estimated based on prior clinical data.” Could the authors clarify what these Bayesian priors were used for? Please, also specify the actual priors employed and run a sensitivity check.

Response: We understand the question of the reviewer concerning the Bayesian priors. To clarify: the Bayesian priors were not used in the estimation of the final models. Rather, they were employed only in the experimental design stage, to construct a Bayesian D-efficient design [4]. The priors reflected expected directions of effects based on existing clinical evidence (e.g., negative for losing independence, positive for life expextancy).We have clarified this in the Methods section (lines 148-151).

6. Comment 6; Interaction terms: The authors include certain interaction terms between personal characteristics and choice attributes in their models. In page 8 they argue that the specific interactions to be tested were chosen based on “literature and consensus of the study team”, without providing further explanation on these statistical design choices. The authors should provide a more detailed justification of these analytical choices, referencing relevant literature.

Response: We apologize for any lack of clarity concerning the interaction terms. The interaction terms included in our models were not related to the experimental design, but to the analytical stage. In DCEs, it is common to test whether preferences for attributes vary systematically by respondent characteristics. We therefore specified interactions between selected socio-demographic variables and attributes, based on both prior evidence in the literature and clinical relevance as judged by our study team. We have clarified in the Methods that these interactions were introduced to capture preference heterogeneity and were pre-specified based on theoretical and empirical grounds, rather than being chosen post hoc (lines 192-195).

7. Comment 7; Introducing latent class analysis: Not every reader will have a solid grasp on how latent class analysis works. The paper would benefit from providing a more extensive introduction to it.

Response: We thank the reviewer for this suggestion. To improve clarity, we have added a more detailed description of the latent class analysis in the Methods section (lines 200-204).

8. Comment 8; Unit of analysis latent class analysis: When running the latent class analysis, was the unit of analysis the individual or the choice? The text seems to suggest that individuals were classified into two groups, but I am not convinced. If the unit of analysis is indeed the individual, and N=333 (as reported in Table 2), then there cannot be a group with exactly 22.4% of individuals, since this would correspond to 74.5 individuals. The group should consist of either 74 individuals (22.2% of the sample) or 75 individuals (22.5%). Could the authors clarify how the multiple observations per individual were handled in this analysis and how group percentages were derived?

Response: We appreciate the reviewer’s careful observation. In latent class discrete choice models, the unit of analysis is the choice, but class membership is estimated at the respondent level. Each respondent contributes multiple choices to the likelihood, which is then used to calculate posterior probabilities of belonging to each latent class. Class sizes are reported as the sum of these posterior probabilities across all respondents, which results in expected proportions such as 22.4% rather than exact integers. In other words, the percentages represent the average probability of respondents belonging to a class, and therefore fractions of individuals are possible.

9. Comment 9; non-monotonic life expectancy preferences: What do the authors make of the non-monotonic life expectancy preferences? (the fact that people assign a higher utility to 2 additional life expectancy years than to 5 years). What could explain these findings? I think the authors should elaborate on this further in the discussion.

Response: To address this fair point made by the reviewer, we have expanded the Discussion section (lines 304-309). Specifically, we now highlight that while an increase of two years in life expectancy was found to be relevant for preferences, an increase of five years did not demonstrate the same influence. As noted in conversations with participants, a possible explanation may lie in the perceived quality of the additional years. Participants may value a shorter life expectancy more highly if the extra years are expected to be accompanied by less desirable health outcomes or other unfavorable treatment attributes.

Reviewer 2

1. Comment 1; Language and clarity: manuscript would benefit from careful proofreading and language editing to improve readability and ensure clarity of expression.

Response: The manuscript has been read through carefully once more, and wording has been adjusted where necessary to improve clarity and overall readability.

2. Comment 2; Presentation of Logit estimates: The current tables report coefficients without clearly indicating their interpretation. I recommend presenting average marginal effects, which would make the results more interpretable and policy-relevant.

Response: We thank the reviewer for this suggestion. We agree that average marginal effects (AMEs) can in some cases aid interpretation. However, in discrete choice experiments the primary outcome of interest is the set of preference weights (β’s), which directly quantify the relative utilities of attribute levels. These coefficients form the basis from which all secondary quantities — including predicted probabilities, willingness-to-pay estimates, and marginal effects — can be derived [3, 4]. AMEs in contrast are conditional on the specific distributions of covariates and the scenarios under consideration, and therefore less generalizable beyond the exact design context. For this reason, we report coefficients as the main results, consistent with established methodological standards, while noting in the text how these can be interpreted in applied settings

3. Comment 3; Sample representativeness: While the Netherlands is indeed experiencing an aging process, the sample appears to cover a specific geographic area and may not be representative of the broader population. The authors should discuss potential sample selection issues more transparently and clarify the limitations of generalizing the results.

Response: We acknowledge the importance of sample representativeness, however, we would like to clarify that there is no indication that this study disproportionately attracted participants from a specific geographic area. The sample was recruited from a national fair, located in the center of the Netherlands, for people over 50 years of age, which is known to attract attendees from all over the country. Based on the background information of the fair we do not expect systematic geographic selection effects.

[1] Orme B. Sample size issues for conjoint analysis studies. Sequim: Sawtooth software technical paper 1998.

[2] Johnson R, Orme B, Software S. Getting the Most from CBC.

[3] Hauber AB, González JM, Groothuis-Oudshoorn CG, Prior T, Marshall DA, Cunningham C et al. Statistical methods for the analysis of discrete choice experiments: a report of the ISPOR conjoint analysis good research practices task force. Value in health 2016;19:300–15.

[4] de Bekker-Grob EW, Donkers B, Jonker MF, Stolk EA. Sample size requirements for discrete-choice experiments in healthcare: a practical guide. The Patient-Patient-Centered Outcomes Research 2015;8:373–84.

[5] McFadden D. Conditional logit analysis of qualitative choice behavior 1972.

---

## [Decision Letter · Decision Letter 1]

19 Oct 2025

What matters most to older adults in treatment decision making: a discrete choice experiment.

PONE-D-25-37657R1

Dear Dr. Hanewinkel,

We’re pleased to inform you that your manuscript has been judged scientifically suitable for publication and will be formally accepted for publication once it meets all outstanding technical requirements.

Kind regards,

José Alberto Molina

Academic Editor

PLOS ONE

Additional Editor Comments (optional):

Reviewers' comments:

Reviewer's Responses to Questions

**Comments to the Author**

Reviewer #1: All comments have been addressed

Reviewer #2: (No Response)

2. Is the manuscript technically sound, and do the data support the conclusions?

Reviewer #1: Yes

Reviewer #2: (No Response)

3. Has the statistical analysis been performed appropriately and rigorously?

Reviewer #1: Yes

Reviewer #2: (No Response)

4. Have the authors made all data underlying the findings in their manuscript fully available?

Reviewer #1: Yes

Reviewer #2: (No Response)

5. Is the manuscript presented in an intelligible fashion and written in standard English?

Reviewer #1: Yes

Reviewer #2: (No Response)

Reviewer #1: All comments and remaining concerns have been adequately addressed. I congratulate the authors on their work.

Reviewer #2: While I appreciate the authors’ efforts to revise the manuscript, the key concerns raised in my previous report have not been satisfactorily addressed. As a result, the paper still suffers from the same fundamental problems that limit its contribution and rigor.

**Do you want your identity to be public for this peer review?** For information about this choice, including consent withdrawal, please see our Privacy Policy

Reviewer #1: No

Reviewer #2: No

---

## [Editor Report · Acceptance letter]

PONE-D-25-37657R1

PLOS ONE

Dear Dr. Hanewinkel,

I'm pleased to inform you that your manuscript has been deemed suitable for publication in PLOS ONE. Congratulations! Your manuscript is now being handed over to our production team.

Kind regards,

on behalf of

Professor José Alberto Molina

Academic Editor

PLOS ONE